# Evaluation of the Biocontrol Efficiency of *Bacillus subtilis* Wettable Powder on Pepper Root Rot Caused by *Fusarium solani*

**DOI:** 10.3390/pathogens12020225

**Published:** 2023-01-31

**Authors:** Junqing Qiao, Rongsheng Zhang, Yongfeng Liu, Youzhou Liu

**Affiliations:** Institute of Plant Protection, Jiangsu Academy of Agricultural Sciences, Nanjing 210014, China

**Keywords:** *B. subtilis* PTS-394 wettable powder, pepper root rot, lipopeptides, colonization, biocontrol efficiency

## Abstract

The plant-growth-promoting rhizobacteria (PGPR) *B. subtilis* PTS-394 has been utilized as a biocontrol agent (in a wettable powder form) due to its excellent ability to suppress tomato soil-borne diseases caused by *Fusarium oxysporum* and *Ralstonia solanacearum*. In this study, we evaluated the biocontrol efficiency of *Bacillus subtilis* PTS-394 wettable powder on pepper root rot in pot experiments and field trials. *B. subtilis* PTS-394 and its lipopeptide crude extract possessed excellent inhibition activity against *Fusarium solani*, causing pepper root rot; in an antifungal activity test *B. subtilis* PTS-394 wettable powder exhibited a good ability to promote pepper seed germination and plant height. The experiments in pots and the field indicated that *B. subtilis* PTS-394 wettable powder had an excellent control effect at 100-fold dilution, and its biocontrol efficacy reached 69.63% and 74.43%, respectively. In this study, the biocontrol properties of *B. subtilis* PTS-394 wettable powder on pepper root rot were evaluated and its application method was established. It was concluded that B. subtilis PTS-394 wettable powder is a potential biocontrol agent with an excellent efficiency against pepper root rot.

## 1. Introduction

Pepper (*Capsicum annuum* L.) is the most popular vegetable with high nutrient contents and vitamins A and C. According to the FAO, China is the largest producer of peppers and the plant area of green pepper is 754,718 hectares in 2021 [1]. Facility cultivation is a common approach to growing pepper in Jiangsu Province, China with an annual planting area of 100,000 hm^2^ [2]. However, the continuous cropping of pepper results in decreased yield, soil-borne plant pathogen accumulation, and soil microbial community disruption. Pepper soil-borne disease occurrence from 2016 to 2017 showed that pepper root rot is the main soil-borne disease [2]. Currently, pepper root rot is commonly controlled by chemical pesticides. However, excessive chemical pesticide use leads to environmental pesticide residues and pathogen resistance. The deployment of biocontrol micro-organisms has been attempted to control plant diseases and promote plant growth [3]. *Bacillus*, *Pseudomonas*, and *Trichoderma* species are the main biocontrol microbial resources [4,5].

The plant rhizosphere is rich in nutrients due to the accumulation of various plant exudates and is inhabited by beneficial and pathogenic micro-organisms. Beneficial bacteria, termed plant-growth-promoting rhizobacteria (PGPR), can promote plant growth and suppress soil-borne plant pathogens when they colonize the rhizosphere and root surface or grow within radicular tissues [6]. Some studies have proven that the successful colonization of PGPR in the plant rhizosphere is the first prerequisite step [7,8]. During colonization, PGPR strains produce a wide spectrum of bioactive metabolites such as antibiotics, siderophores, volatile organic compounds (VOCs), and quorum-sensing signals to compete with other micro-organisms and survive in the rhizosphere [9,10]. Among these PGPR strains, the *Bacillus* species has received extensive attention and has become an important potential resource for biofertilizers or biopesticides, which are frequently used to control soil-borne plant diseases in the field and greenhouse [4,7].

Recently, many investigations on the control of pepper diseases and the growth promotion of pepper by the *Bacillus* species had been conducted. It was reported that anti-oomycete *Bacillus amyloliquefaciens* and *Bacillus velezensis* exerted a combined effect by directly antagonizing the pathogen and enhancing the pepper resistance to control *Phytophthora capsica* [11]. *B. velezensis* 2A-2B had the capacity to inhibit the growth of *Phytophthora capsici, Fusarium solani, Fusarium oxysporum,* and *Rhizoctonia solani,* and induced the expression of *NPR1*, a key gene controlling local resistance and systemic acquired resistance in plant immunity [12]. *Bacillus subtilis* SL-44 could not only induce systemic resistance of pepper seedling control wilt disease but also produce antifungal compounds to inhibit the growth of *R. solani* [13]. Additionally, the root colonization by *B. subtilis* SL-44 had the ability to promote pepper plant growth [14]. Cisternas Jamet et al. [15] showed that *B. amyloliquefaciens* affected the biochemical composition of fruits, and changed the content of calcium, iron, and vitamin C. Ying-Ru Liang et al. [16] also reported that the colonization of *B. subtilis* Ydj3 on pepper root could increase the yield, fruit weight, and vitamin C content significantly compared with those of the control. *Bacillus* strains could not only control plant diseases by colonizing the rhizosphere, competing with pathogens for space and nutrition, inhibiting pathogens’ growth and expansion by secreting lipopeptides, and stimulating plant-induced systemic resistance [17,18,19,20], but also increase the growth and yields of agricultural crops through the production of indole-3-acetic acid (IAA), gibberellin acid (GA), siderophores, and phosphorous solubilization [21,22].

*Bacillus subtilis* PTS-394, a PGPR, was isolated from the tomato rhizosphere and was fully sequenced in 2014 [23]. It has been shown to suppress tomato soil-borne diseases caused by *Fusarium oxysporum* and *Ralstonia solanacearum* [24]. *B. subtilis* PTS-394 can colonize the tomato rhizosphere, produce a variety of extra-cellular proteolytic enzymes, siderophores, and lipopeptides, and induce tomato disease resistance [24,25]. *B. subtilis* PTS-394 has been developed into a biocontrol agent (wettable powder) after fermentation medium optimization [26].

In this study, the *B. subtilis* PTS-394 wettable powder biocontrol efficiency on pepper root rot was evaluated by the disc diffusion method, mass spectrometry, greenhouse pot experiments, and field trials. This study has established the suitable application dose parameters and the time of use in the field of *B. subtilis* PTS-394 wettable powder to control pepper root rot. It has also expanded on the biocontrol targets of *B. subtilis* PTS-394 wettable powder and provided the theoretical basis and technical information for its application in the field.

## 2. Results

### 2.1. Antifungal Activity of B. subtilis PTS-394 and Its Wettable Powder

The inhibitory effects of the PTS-394 lipopeptide crude extract grown in a YPG medium and the PTS-394 wettable powder on *F. solani* spore germination and mycelial growth were tested by the disc diffusion method (Figure 1). The inhibition zone of the lipopeptide crude extract from PTS-394 wettable powder on mycelium growth was 5 mm, and the spore germination inhibition zone diameter was 13.33 mm. The inhibition zone of the lipopeptide crude extract of PTS-394 grown in YPG medium against mycelia was 8.67 mm, and the diameter of the inhibition zone against spore germination was 20.67 mm. Thus, the inhibitory effect on spore germination and mycelial growth of the lipopeptide extract of *B. subtilis* PTS-394 wettable powder was the same as PTS-394 grown in the YPG medium. However, its inhibition effect was significantly lower than that of PTS-394 grown in YPG, which might be caused by the loss of lipopeptides during the wettable powder production and processing.

### 2.2. Identification of Lipopeptide from B. subtilis PTS-394 and Its Wettable Powder

Matrix-assisted laser desorption/ionization-time of flight mass spectrometry (MAL-DI-TOF-MS) was used to identify the lipopeptides extracted from *B. subtilis* PTS-394 and its wettable powder. Both the crude extract of PTS-394 grown on a YPG medium and the wettable powder contained three types of lipopeptides, Surfactin, Iturin, and Fengcyin (Table 1). Surfactin with molecular weights (MW) of 1021.6, 1035.7, and 1049.6, Iturin with MW of 1057.5, and Fengycin with MW of 1490.8 were detected in the PTS-394 wettable powder and PTS-394 grown in a YPG medium (Table 1). In addition, the extract of PTS-394 grown in a YPG medium also contained Surfactin with MW of 993.6 and 1007.6, Iturin with MW of 1043.5 and 1085.6, and Fengycin with MW of 1476.8 and 1504.8.

### 2.3. Growth-Promoting Effect of B. subtilis PTS-394 Wettable Powder on Pepper

The seed soaking experiment germination results showed that different dilutions of PTS-394 wettable powder had different effects on pepper seed germination (Table 2). The germination rate of the sterilized water control was 71.11%, while the seed germination rate of the PTS-394 powder in 1:100 and 1:1000 dilutions treatments was 80.00% and 81.11%, respectively, significantly promoting more seed germination than the control. However, the 1:10 dilution treatment resulted in only a 46.67% seed germination rate, significantly inhibiting the germination of pepper seeds.

In addition, the pepper seedling germination rates in pot experiments were 72.22% and 75.56% after 8 days of application of PTS-394 wettable powder in soil at a volume ratio of 1:100 and 1:1000, respectively. The germination rate of the control treatment was 63.33%. The plant height of pepper plants in pots treated with PTS-394 powder at 1:1000 and 1:100 dilutions was 20.75 cm and 21.21 cm, respectively, while the plant height of the control was only 17.54 cm. This illustrates the growth-promoting effects of PTS-394 powder on plant height, as it was increased by 18.30% and 20.92%, respectively. These results indicated that the lower concentration of PTS-394 powder significantly promoted pepper seed germination and plant height.

### 2.4. B. subtilis PTS-394 Wettable Powder Biocontrol Effect on Pepper Root Rot in Pot Experiments

The disease index of all treatments was investigated after transplantation and the control effect was calculated. The disease index of PTS-394 wettable powder at 100-fold dilution and 1000-fold dilution on pepper root rot was 17.36 ± 1.84% and 44.21 ± 1.60%, respectively. The disease index of water control treatment was 57.18 ± 1.45%. Both treatments of PTS-394 wettable powder were significantly lower than the control treatment. The biocontrol effect of PTS-394 wettable powder at 100-fold dilution on pepper root rot was 69.63 ± 3.19% in the greenhouse pot experiment. In contrast, the control effect of PTS-394 wettable powder diluted 1000-fold was only 22.69 ± 1.03%. The results indicated that the control efficacy gradually decreased with the decrease in the dose applied (Figure 2).

### 2.5. B. subtilis PTS-394 Dynamic Colonization on the Pepper Rhizoplane

The dynamic colonization of PTS-394 on the pepper rhizoplane was detected based on the characteristic PTS-394 red colonies on the YPG agar plates (Figure 3). After treatment with PTS-394 powder at 100-fold dilution, PTS-394 colonization on pepper roots initially decreased before steadily increasing over time. Almost 3.5 × 10^7^ CFU per gram of the root was detected 24 h after inoculation, which dropped to 2 × 10^5^ CFU/g of the root 9 days post-inoculation. After the ninth day, the PTS-394 cell number increased and reached 4.5 × 10^6^ CFU/g of the root at 15 days post-inoculation. In the PTS-394 WP (1:100) + FS treatment, the dynamic curve of PTS-394 colonization on pepper root was the same as that of the treatment without *F. solani* inoculation. The amount of colonization of PTS-394 in the PTS-394 WP (1:100) + FS treatment was significantly higher than that of the treatment PTS-394 at 3, 5, and 7 days after inoculation. At the ninth day, the number of PTS-394 cells was the lowest and then increased at the 15th day.

### 2.6. B. subtilis PTS-394 Wettable Powder Bicontrol of Pepper Root Rot in Field

A field experiment was performed to investigate the biocontrol efficiency of PTS-394 wettable powder on pepper root rot in different dilutions (1:100), (1:200), along with a control (water treated) (Table 3). Firstly, the PTS-394 wettable powder was mixed with the nursery soil in a 1:100 volume ratio to establish strong seedlings. After transplanting, the roots were inoculated with PTS-394 wettable powder at 1:100 and 1:200 dilutions. The disease incidence was investigated when disease symptoms had appeared and were stable. The pepper root rot incidence rate in the control block was 7.96 ± 0.32%. However, the incidence rate with PTS-394 (1:100) was 2.06 ± 0.28%, with a biocontrol efficacy of 74.44 ± 0.96%, and the incidence rate with PTS-394 (1:200) was 2.50 ± 0.16%, with a biocontrol efficacy of 68.57 ± 3.78%. Notably, we observed that pepper fruits were larger in the PTS-394 treatment than in control.

## 3. Discussion

*Bacillus* species have been developed as biofertilizers and biopesticides due to their ability to control soil-borne diseases and promote plant growth [4]. The biocontrol activities include several mechanisms, such as antagonism, competition, and plant resistance induction [6,27]. Lipopeptides produced by *Bacillus* play a major role in the antagonistic activity to pathogens and colonization competition [20]. Bais et al. [8] demonstrated that Surfactin absence resulted in the decrease of biofilm formation and thereby reduced its ability to colonize the roots, which led to a decrease in the control effect of *Pseudomonas syringae*. Fengycin, secreted by *B. subtilis* ZD01, effectively limited the development of the lesion area of potato early blight and caused mycelial and conidial malformations [28]. Zhao et al. [29] showed that fengycin impacted the cell membrane of *F. graminearum* and led to the release of cell contents. In addition, *Bacillus* and its lipopeptides also played an important role in the inducing of plant-defense-related enzymes and expression of defense-related genes including phenylalanine ammonia lyase (PAL), peroxidase (POD), polyphenol oxidase (PPO), PR proteins, and *NPR1* gene [13,20]. *B. subtilis* SL-44 induced the systemic resistance of pepper seedling against wilt disease by jasmonic-acid-dependent signaling pathway [13]. Ongena et al. determined Surfactin and fengycin of *Bacillus subtilis* to be elicitors of induced systemic resistance in plants [30]. Previous studies indicated that *B. subtilis* PTS-394 had a capacity to induce the systemic broad-spectrum disease resistance in tomato leaf by salicylic acid (SA)-dependent signaling pathway, and increased the plant defense enzymes (PAL, PPO, POX, and Lox) [24].

The *B. subtilis* PTS-394 genome contains gene clusters of Non-ribosomal Peptide Synthetase Gene (NRPS), which respond to lipopeptide synthesis [23]. In this study, the MALDI-TOF-MS results showed that the lipopeptides Surfactin, Iturin, and Fengycin were detected in both the YPG medium culture and the PTS-394 wettable powder crude extract, which had variable antagonistic spectra to different fungal genera. Surfactin contributed significantly to the antagonistic activity against *F. solani*, *Rhizoctonia solani*, and *Sclerotinia sclerotiorum* [31]. The plate antagonistic activity results showed that the crude lipopeptide antibiotic extract in *B. subtilis* PTS-394 WP still strongly inhibits *F. solani* mycelial growth and spore germination. Although three lipopeptides were detected in *B. subtilis* PTS-394 Wettable powder, the content, the isoform number, and the MS detection intensity were lower than in *B. subtilis* PTS-394 YPG medium culture. The MS results revealed that Surfacin was the major compound in *B. subtilis* PTS-394 wettable powder. Therefore, we proposed that *B. subtilis* PTS-394 wettable powder exerted the control effect on pepper root rot not only by the antagonistic activity of lipopeptides against *F. solani* but also by the inducing of the plant immunity.

The present study showed that PTS-394 wettable powder had a significant promotion effect on seed germination and seedling plant height. Several works of research revealed that *bacillus* strains had the ability to promote pepper plant growth in seed germination, plant height, fruit weight, and vitamin C, and change the biochemical composition of fruits. Here, it would be suggested that PTS-394 wettable powder could be used at low doses as a growth-promoting fertilizer during the fruit expansion period, which may promote yield and simultaneously improve the content of fruit, such as increased vitamin C content.

The plant rhizosphere is enriched by many nutrients and is the habitat of various beneficial or harmful micro-organisms [6]. The colonization of biocontrol agents is a competitive process with indigenous micro-organisms in ecological niches. Herein, the superior colonization competence of *Bacillus* species on the rhizosphere is crucial for their excellent biocontrol properties [7,32]. Previous studies showed that *B. subtilis* PTS-394 could colonize tomato roots, temporarily affect indigenous bacteria and fungi on the rhizosphere, and significantly suppress the population of *Fusarium* species in the soil [7]. In our study, we found that PTS-394 could colonize the pepper root surface with or without *F. solani*, and its colonization profile was similar to that on the tomato root surface [7]. A previous study had also revealed that *B. subtilis* PTS-394 had a transient effect on the populations of bacteria, fungi, and actinomycetes in the pepper rhizosphere [33]. Therefore, these results indicate that *B. subtilis* PTS-394 can successfully colonize plant roots and establish the foundation for its biocontrol and growth promotion properties.

Pepper root rot caused by *F. solani* has seriously affected pepper yield in Jiangsu province. Applying chemical pesticides to control soil-borne diseases is known to result in pesticide residues and soil and water pollution. With the extensive research on PGPR, the application of beneficial micro-organism products to control plant soil-borne diseases has become an efficient approach. For example, *Bacillus subtilis* B99-2 microencapsulation was used to control tomato blight [34], *Bacillus amyloliquefaciens* 1619 water-dispersible granules were applied for the control of tomato wilt and root-knot nematodes [35,36], and *Bacillus subtilis* Bs916 was applied for the control of rice sheath blight and tomato wilt [37,38]. Zhang et al. [39] reported that the application of a consortium of PGPR strains could reduce the sweet pepper disease, with the improvement of fruit quality and soil properties. *Bacillus subtilis* XZ18-3 was developed into a wettable powder with a control efficacy of 88.28% against *Rhizoctonia cerealis* on wheat [40]. In this work, *B. subtilis* PTS-394 wettable powder has an excellent and stable biocontrol effect both in pot and field trials and is a potential biocontrol agent for controlling the pepper root rot.

## 4. Conclusions

In this study, the *B. subtilis* PTS-394 wettable powder biocontrol properties on pepper root rot were evaluated. The dosage and its application method in the field were established. It was concluded that the control effect of PTS-394 WP against root rot of pepper was up to 74.43% at a 100-fold dilution dosage. Taken together, *B. subtilis* PTS-394 wettable powder has an excellent efficiency on the controlling of pepper root rot and is a potential biocontrol agent.

## 5. Materials and Methods

### 5.1. Bacterial Strains and Growth Conditions

*B. subtilis* PTS-394 and the pepper root rot pathogen, *F. solani* HALJ3-1, were isolated from pepper root and preserved by our laboratory. The biocontrol agent *B. subtilis* PTS-394 in a wettable powder (WP) formulation containing 2 × 10^10^ CFU spore/g was produced by Yang-zhou Luyuan Biochemical Co., Ltd., located in Yangzhou city, China. The *Bacillus* strains were grown in a Yeast Peptone Glucose medium containing 0.5% yeast extract, 0.5% peptone, and 0.5% glucose (YPG). The strain *F. solani* HALJ3-1 was cultured at 28 °C on a Potato Dextrose Agar (PDA) medium containing potato 200 g/L, glucose 20 g/L, and agar 15 g/L.

### 5.2. Preparation of the Lipopeptides Crude Extract

A single PTS-394 colony was cultured in a tube containing 5 mL of YPG medium at 30 °C, 200 rpm for 12 h. Then, 1 mL from this culture was transferred to 100 mL YPG medium in a 500 mL flask, followed by incubation for 48 h at 30 °C. The supernatant was collected by centrifugation at 6000 rpm for 10 min at 4 °C. Then, 6 mol/L HCl was added to adjust the pH to 2.0 and stored at 4 °C overnight. The acid precipitate was extracted with methanol and neutralized for 12 h. The crude extract was obtained by sample suspension in methanol and subsequent filtration (0.22 µm, Nylon). The crude extract was then stored at −20 °C. At the same time, the lipopeptide crude extract was isolated from the *B. subtilis* PTS-394 wettable powder with methanol (5 g powder with 5 mL methanol) and filtered with a 0.22 μm, Nylon filter.

### 5.3. Antifungal Activity Assay

The antifungal activity of *B. subtilis* PTS-394 and the lipopeptide crude extract were determined by the disc diffusion method on PDA plate with some modification [41,42]. The inhibitory activity on mycelium was evaluated as follows: *F. solani* HALJ3-1 was grown on PDA plates at 28 °C for three days in darkness. The fungi pellets were punched along the edge of the colonies in 5 mm diameter and then placed at the center of a PDA plate. For antagonistic activity of strain PTS-394, 25 mm away from the pellets, in a “cross” pattern, 1 μL of PTS-394 culture broth was added and YPG medium as a control treatment. For inhibition of lipopeptide crude extract, 25 mm away from the pellets, a hole was punched in a “cross” pattern, and 50 μL of lipopeptide extract was added, with 50 μL methanol as a control treatment. Each treatment was repeated three times. All plates were incubated at 28 °C for 4 d, and the inhibition zone was investigated. The inhibitory activity on spore germination was evaluated as follows: the fungal pellets (5 mm) were collected and washed in sterilized water, then the spore number was adjusted at 10^4^ CFU/mL, and 100 μL was plated on PDA agar. Then, 25 mm from the center of the plate, a hole was punched in a “cross” pattern, and 20 μL of lipopeptide extract was added. All plates were incubated at 28 °C for 4 d, and the inhibition zone was measured. Each treatment was repeated three times, along with 20 μL methanol as a control treatment.

### 5.4. Identification of Lipopeptides with MALDI-TOF-MS

The lipopeptide crude extract from PTS-394, grown in a YPG medium, and from the PTS-394 wettable powder was assessed using matrix-assisted laser desorption/ionization-time of flight mass spectrometry (MAL-DI-TOF-MS) in the 200–2000 nm wavelength range. One μL of crude extracts was analyzed in a MALDI-TOF instrument (Bruker Daltonics Inc., Brerica, MA, USA) containing a 337 nm nitrogen laser for desorption and ionization [43]. α-Cyano-4-hydroxycinnamic acid was used as the matrix.

### 5.5. Growth-Promoting Effects of B. subtilis PTS-394 Wettable Powder on Pepper

*B. subtilis* PTS-394 wettable powder was serially diluted for the pepper seed germination experiment at a 1:10, 1:100, and 1:1000 ratio. Pepper seeds were sterilized by soaking in 3% sodium hypochlorite solution for 5 min and washed with sterile water 3 times. The sterilized seeds were then soaked in different dilutions of PTS-394 solution for 24 h. The control seeds were soaked in sterile water. Subsequently, the seeds were removed, dried from excess solution, and placed in Petri dishes containing 4 layers of sterile filter paper and covered with 2–3 layers of sterilized gauze and 10 mL of sterilized water. Each plate contained 30 seeds and was repeated 3 times. The germination rate was measured after 8 days.

*B. subtilis* PTS-394 wettable powder was mixed, at 1:100 and 1:1000 times volume ratio, with nursery soil containing a mixture of field soil, organic manure, and vermiculite (1:1:1, *v*/*v*). Pepper seeds were sown into the soil, and a control treatment with no PTS-394 wettable powder was added. They were placed in a greenhouse (natural light, 15–28 °C) and were well watered. Each treatment contained 30 seeds and was repeated three times. The seedlings were evaluated after 8 d, and the germination rate was calculated.

To evaluate the plant-growth-promoting properties of *B. subtilis* PTS-394 wettable powder, nursery soil containing a mixture of field soil, organic manure, and vermiculite (1:1:1, *v*/*v*) was prepared for a greenhouse pot experiment. The seeds were sown into the soil, and when the seedlings reached the 4-leaf stage, the seedlings were transplanted into individual pots in the greenhouse (natural light, 15–28 °C). Three treatments were applied, namely, control (CK), root irrigation with PTS-394 powder at 1:100 dilution (PTS-1:100), and with PTS-394 powder at 1:1000 dilution (PTS-1:1000). In these PTS-394 treatments, each pepper plant was irrigated with 20 mL of PTS-394 powder dilution. Each treatment was performed on 30 plants, and the whole experiment was repeated three times. On the 25th day after transplantation, the height of 10 plants was measured in each replicate, and the *B. subtilis* PTS-394 powder growth-promoting effects were statistically analyzed.

### 5.6. The Colonization of B. subtilis PTS-394 Wettable Powder on the Pepper Rhizoplane

The dynamic colonization of *B. subtilis* PTS-394 wettable powder on the pepper rhizoplane was investigated in pot experiments. The *F.-solani*-inoculated soil was prepared as follows: Firstly, *F. solani* HALJ3-1 was cultured in a PDA medium in darkness. After 5 d of incubation at 28 °C, it was inoculated into a sand medium containing corn flour and sand (2:1 *v*/*v*) in darkness. After 20 d of incubation at 28 °C, the pathogen-inoculated soils were prepared with a 1:1:1:1 volume ratio of the sand medium, field soil, organic manure, and vermiculite for greenhouse experiments.

In the colonization experiment, two treatments were applied: PTS-394 WP (1:100) and PTS-394 WP (1:100) + FS (abbreviation of *F. solani*). *B. subtilis* PTS-394 wettable powder was diluted in a 1:100 ratio with water. Pepper seeds were surface-disinfected and germinated in a nursery soil containing a mixture of vermiculite, field soil, and organic manure (1:1:1, *v*/*v*). Seedlings at the 4-leaf stage were transplanted into pots with the pathogen-inoculated soil or the standard nursery soil, to which 20 mL of PTS-394 powder dilution was added each. Each treatment contained 50 plants and was cultured in a greenhouse under natural lighting with the temperature ranging from 18–30 °C. Pepper root samples were collected from treatments of PTS-394 WP (1:100) and PTS-394 WP (1:100) + FS at 1, 3, 5, 7, 9, 15, and 25 days after transplantation. At the time of sampling, a total of two pepper seedlings were collected per treatment and replicated 3 times. The soil on the root of two pepper plants was removed as clean as possible, and the root was washed with ultrasonic waves for 20 min at a ratio of 10 mL sterile water per gram of root. Afterwards, the soil solution was diluted 10^4^ or 10^5^ times and plated on YPG agar petri dish to count the number of PTS-394 colonies. Then, the amount of PTS-394 per gram of root was calculated. Statistical analysis was performed on the three replicates.

### 5.7. The Biocontrol Effects of B. subtilis PTS-394 Wettable Powder in Pots

The biocontrol effect of *B. subtilis* PTS-394 wettable powder on the pepper root rot was evaluated in pot experiments. Three treatments were performed to evaluate the PTS-394 wettable powder biocontrol properties as follows: PTS-394 WP (1:100) +FS, PTS-394 WP (1:1000) +FS, and FS. Each treatment was performed on 32 plants, and the whole experiment was repeated three times in a completely randomized block design. *B. subtilis* PTS-394 wettable powder was diluted in a 1:100 and 1:1000 ratio with water. The pepper seedlings, pathogen-inoculated, and the control nursery soil were prepared according to the above description. Seedlings at the 4-leaf stage were transplanted into pots with the pathogen-inoculated soil or the standard nursery soil, and watered with 20 mL of PTS-394 powder solution each. Cultivation was continued in a greenhouse under natural lighting with the temperature ranging from 18–30 °C. The disease index was investigated 21 days after transplantation as a proxy for the severity of the root rot. Disease index corresponded to 9 grades as follows: grade 0, the plants are not wilting; grade 1, a few leaves showing recoverable wilting; grade 3, 10% to 30% of leaves wilting, leaves are still green; grade 5, 30.1~50% of leaves obviously wilting, leaves are yellow; grade 7, plants wilting, 50.1~80% of leaves are yellowing and wilting, rhizome browning; grade 9, the whole plants die, roots rot.
DSI (%) = [sum (class frequency × score of rating class)] / [(total number of plants) × (maximal disease index)] × 100
Disease control effect (%) = (Control DSI − Treatment DSI) / Control DSI × 100

### 5.8. The Biocontrol Effects of B. subtilis PTS-394 Wettable Powder in Field

A field experiment was performed at a vegetable greenhouse with serious pepper root rot disease located in Huai’an city (33°41′ N, 118°53′ E), Jiangsu Province, China. The biocontrol efficiency of PTS-394 wettable powder on pepper root rot in different dilutions (1:100), (1:200), along with a control (water-treated) treatment were investigated. Three replicate blocks per treatment were arranged in a randomized complete block design and contained 360 plants per replication. Firstly, *B. subtilis* PTS-394 wettable powder was mixed with the nursery soil to a volume ratio of 1:100 and followed by seed sowing to establish strong seedlings. After the transplantation, *B. subtilis* PTS-394 wettable powder was applied in the rhizosphere according to a dosage of 8 g/plant (1:100) or 4 g/plant (1:200) with 800 mL water. Secondly, 15 days after transplantation, the roots of each seedling were irrigated with 400 mL of *B. subtilis* PTS-394 wettable powder at a dilution of 1:100 or 1:200. During the disease symptom period (about 60 days after transplantation), the disease incidence rate was investigated, and the control effect was analyzed.
Disease incidence rate (%) = the number of diseased plants / the total number of plants × 100
Control efficiency (%) = (incidence rate of control − incidence rate of treatment) / incidence rate of control × 100.

### 5.9. Statistical Analysis

The experimental data were statistically analyzed by SPSS 19.0 software. Duncan’s new complex range method was used to analyze the significance of means at *p* ≤ 0.05 after conducting the analysis of variance (ANOVA) for the data sets from growth promotion and biocontrol experiments of *B. subtilis* PTS-394 wettable powder. Independent two-sample *t*-test was performed in the significantly analysis of the dynamic colonization of *B. subtilis* PTS-394 on the pepper root surface.

## Figures and Tables

**Figure 1 pathogens-12-00225-f001:**
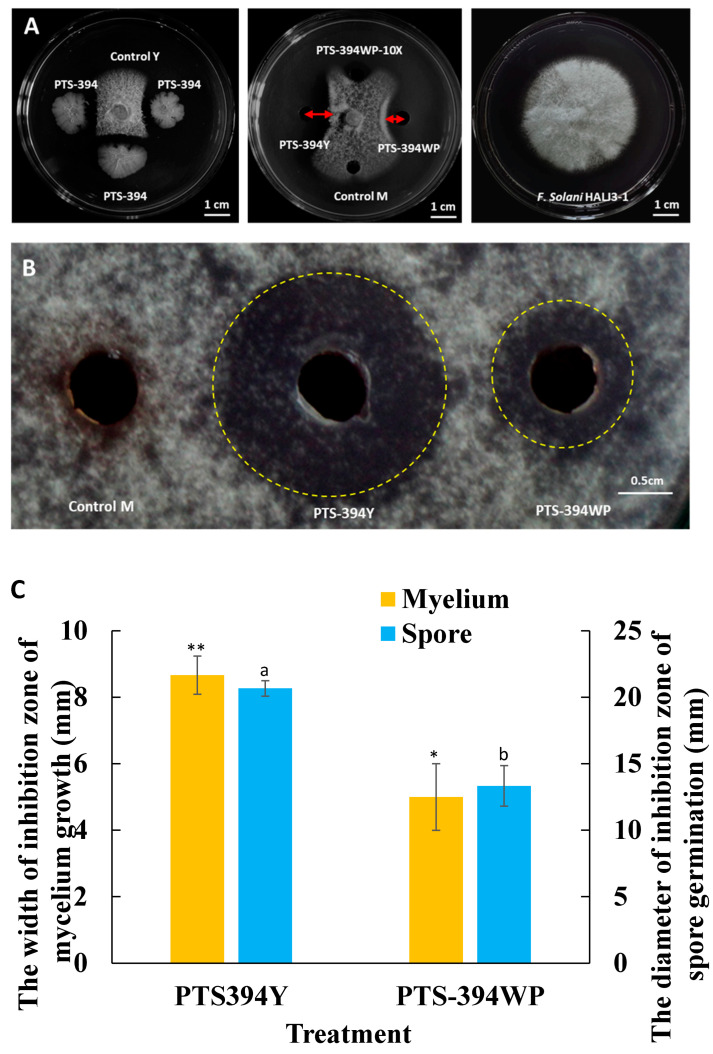
Antifungal activity of *B. subtilis* PTS-394 and its wettable powder on mycelial growth (**A**) and spore germination (**B**) and the width of inhibition zone of mycelium growth and spore germination (**C**). PTS-394 means 1 μL of PTS-394 YPG culture broth; Control Y means YPG medium; PTS-394Y means PTS-394 lipopeptide crude extract grown in a YPG; PTS-394WP means lipopeptide crude extract from PTS-394 wettable powder; PTS-394WP-10 × means lipopeptide crude extract diluted 10 times from PTS-394 wettable powder; Control M means methanol (methanol is the solvent of the crude extract of lipopeptides). Error bars indicate the standard deviation calculated from three independent samples. Different letters and stars indicate significant difference at 0.05 level by Duncan’s new multiple range test.

**Figure 2 pathogens-12-00225-f002:**
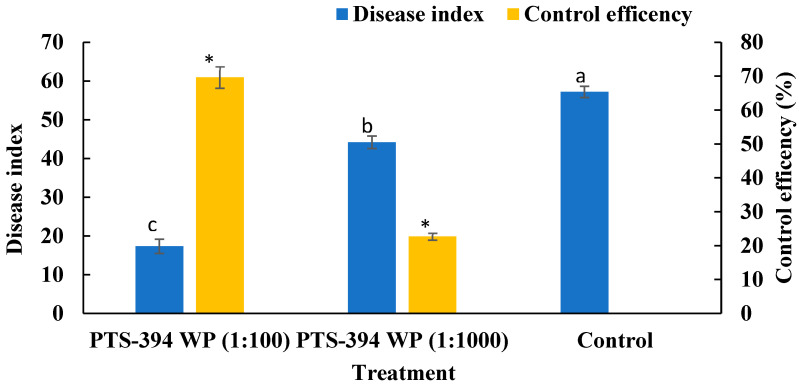
The disease index and control efficiency of *B. subtilis* PTS-394 WP on pepper root rot in pot experiment. Different letters and stars indicate significant difference at 0.05 level by Duncan’s new multiple range test.

**Figure 3 pathogens-12-00225-f003:**
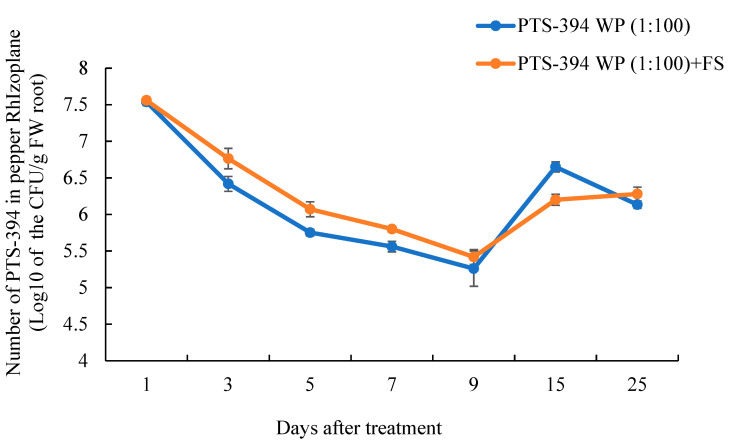
Colonization dynamics of *B. subtilis* PTS-394 on pepper root surface in pot experiments. PTS-394 WP (1:100) means *B. subtillis* PTS-394 wettable powder treatment, and PTS-394 WP (1:100) + FS means *B. subtillis* PTS-394 wettable powder with *F.solani* treatment.

**Table 1 pathogens-12-00225-t001:** Lipopeptides of *B. subtilis* PTS-394 and wettable powder detected by MALDI-TOF-MS.

Mass Peaks (M/Z)	Products	Mass Peaks in PTS-394 YPG M/Z (Intensity) [M + H, Na, K]^+^	Mass Peaks in PTS-394WP M/Z (Intensity) [M + H, Na, K]^+^
Surfactin			
993.6	C13 Surfactin B	1016.6 (15,521.25) [M + Na]^+^	-
1007.6	C13 Surfactin A C14 Surfactin B C13 Surfactin C	1030.6 (84,069.20)/1046.6 (27,524.07) [M + Na, K]^+^	-
1021.6	C14 Surfactin A C15 Surfactin B C14 Surfactin C	1044.6 (110,886.84)/1060.6 (44,451.90) [M + Na, K]^+^	1044.6 (356.55) [M + Na] ^+^
1035.7	C15 Surfactin A C16 Surfactin B C15 Surfactin C	1058.7 (121,828.87)/1074.6 (67,291.75) [M + Na, K]^+^	1058.7 (910.61)/1074.6 (408.72) [M + Na, K]^+^
1049.6	C16 Surfactin A C16 Surfactin C	1072.7 (25,248.00)/1088.7 (7889.02) [M + Na, K]^+^	1088.7 (382.82) [M + K]^+^
Iturin			
1043.5	C14 Iturin B	1066.7 (6712.2) [M + Na]^+^	-
1057.5	C15 Iturin B	1080.7 (9859.62) [M + Na]^+^	1096.5 (339.58) [M + Na]^+^
1085.6	C17 Iturin B	1086.7 (6015.7) [M + H]^+^	-
Fengycin			
1476.8	C15 Fengycin B C17 Fengycin A	1499.8(18,130.15) [M + Na ]^+^	-
1490.8	C18 Fengycin A C16 Fengycin B	1513.8 (21,506.17)/1529.8 (6258.52) [M + Na, K]^+^	1491.8 (221.47) [M + H]^+^
1504.8	C17 Fengycin B	1527.8 (18,134.60)/1543.8 (5152.01) [M + Na, K]^+^	-

“-”, means no detection.

**Table 2 pathogens-12-00225-t002:** The promotion effect of *B. subtilis* PTS-394 WP on pepper.

Treatment	Germination Percentage on Plate (%)	Germination Percentage in Pot (%)	Height of Plant (cm)
PTS-394 WP 1:10	46.67 ± 3.33 c	No	No
PTS-394 WP 1:100	80.00 ± 3.33 a	72.22 ± 3.85 a	21.21 ± 2.38 a
PTS-394 WP 1:1000	81.11 ± 5.09 a	75.56 ± 1.92 a	20.75 ± 2.59 a
Control	71.11 ± 1.92 b	63.33 ± 3.33 b	17.54 ± 2.52 b

The data in the table are mean ± SD. ‘No’ means treatment was not carried out. Different letters in the same column indicate significant difference at 0.05 level by Duncan’s new multiple range test.

**Table 3 pathogens-12-00225-t003:** The control efficiency of *B. subtilis* PTS-394 WP on pepper root rot in field.

Treatment	Incidence of Disease (%)	Control Efficiency (%)
PTS-394 WP 1:100	2.06 ± 0.16 c	74.44 ± 0.96 a
PTS-394 WP 1:200	2.50 ± 0.28 b	68.57 ± 3.78 b
Control	7.96 ±0.32 a	

The data in the table are mean ± SD. ‘No’ means treatment was not carried out. Different letters in the same column indicate significant difference at 0.05 level by Duncan’s new multiple range test.

## Data Availability

Not applicable.

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
