# Peer review of "Evaluation of the Biocontrol Efficiency of Bacillus subtilis Wettable Powder on Pepper Root Rot Caused by Fusarium solani"

_pathogens, 2023, doi:10.3390/pathogens12020225_

Round 1
Reviewer 1 Report
The authors have used an ecofriendly approach for controlling one of the important diseases of pepper in the manuscript titled "Evaluation of the Biocontrol efficiency of Bacillus subtilis wettable powder on Pepper Root Rot caused by Fusarium solani" but I have some queries related to the manuscript.
Comment 1: The authors have mentioned in fig 1. the control plate but it is not justified in figure or in the manuscript what control is used. They should clearly mention the control used, is it a fungicide that is commonly used for the control of this disease that is usually used as a control in these experiments or anything else has been used.
Comment 2: The dimensions or diameter of the control plate has not been mentioned that can provide a comparison about the efficacy of the bio-agent used as biocontrol. The control and biocontrol agent treated plate visually show almost the same growth I advise changing the figure.
Comment 3: For plant fungal cultures usually the Disc diffusion method or food poison method is used for checking the zone of inhibition but nowhere in the manuscript clear name of the standard method is mentioned. The authors must provide more details about the method used for checking the zone of inhibition in this research.
Author Response
Comments and Suggestions for Authors
The authors have used an ecofriendly approach for controlling one of the important diseases of pepper in the manuscript titled "Evaluation of the Biocontrol efficiency of Bacillus subtilis wettable powder on Pepper Root Rot caused by Fusarium solani" but I have some queries related to the manuscript.
Response: Thanks for your suggestion which are very useful to improve the quality of our manuscript. We had revised the manuscript according to your specific comments and labled in red.
Comment 1: The authors have mentioned in fig 1. the control plate but it is not justified in figure or in the manuscript what control is used. They should clearly mention the control used, is it a fungicide that is commonly used for the control of this disease that is usually used as a control in these experiments or anything else has been used.
Response:The Fig1 was modified and separated into 3 parts and the control treatments information were explained in the legend of Fig1.
Comment 2: The dimensions or diameter of the control plate has not been mentioned that can provide a comparison about the efficacy of the bio-agent used as biocontrol. The control and biocontrol agent treated plate visually show almost the same growth I advise changing the figure.
Response: The antifungal activity of B. subtilis PTS-394 and the lipopeptide crude extract were determined in the present study. In the experiment, the YPG medium and the methanol were used as control treatment. Both two treatment showed that there were not inhibition activities on mycelium and spores of F. solani. In order to obvsered clearly, the red arrows and yellow circles were marked; in addition, the scale bar were added in the bottom of Figure 1. We also meseaued the dimensions or diameter of inhibition zone and perfomed histogram analysis(Fig.1 C).
Comment 3: For plant fungal cultures usually the Disc diffusion method or food poison method is used for checking the zone of inhibition but nowhere in the manuscript clear name of the standard method is mentioned. The authors must provide more details about the method used for checking the zone of inhibition in this research.
Response: The discription in section“Antifungal activity assay” was not clear. In this study, we also used the Disc diffusion method with some modification, the experiment details were added in the reviesed as follow. In this study, a hole was used to instead of the filter paper in the nomal Disc diffusion method. Please check the revised section “Antifungal activity assay”.

Reviewer 2 Report
Plese note the yellow bar color in Fig 1, and the key box does not match. Both of them should be yellow.
Author Response
Comments and Suggestions for Authors
Plese note the yellow bar color in Fig 1, and the key box does not match. Both of them should be yellow.
Response: Thanks for your suggestion. The Fig1 was modified and the yellow bar color was matched with the box.
Reviewer 3 Report
Comments and Suggestions for Authors
Manuscript ID: pathogens-2146606
The paper entitled “Evaluation of the Biocontrol efficiency of Bacillus subtilis wettable powder on Pepper Root Rot caused by Fusarium solani” was carefully reviewed. The purpose of this paper is to assess the biocontrol efficiency of B. subtilis PTS-394 wettable powder on pepper root rot in pot experiments and field trials.
This manuscript is fairly well written but needs extensive revision. Here are some critical points:
1. The relevance of the research has not been clearly identified and discussed;
2. Technical clarifications: many deficiencies in the Materials & Methods section compromise the understanding of ideas and concepts and affect the quality of the paper;
3. Poorly prepared discussion lacking appropriate interpretation (out of context!). The Discussion Section should be concise and relevant to the results. Furthermore, some statements are simply incorrect and/or are not supported by any scientific literature.
4. Many "key" references have been missed, and the list of references is not up-to-date;
5. The manuscript is full of typos.
Detailed comments:
Abstract
- Improve on the conclusions of the abstract.
Introduction
- This section still needs to be deeply improved. Enlarge the state of the art by adding other relevant and recent works in the field. Add one or two paragraphs in the Introduction Section to present the studies elucidating the efficacy of Bacillus subtilis against pathogenic fungi in pepper crops.
- Page 1: “Jiangsu Province, …”. Indicate the country.
- Page 1: “with an annual planting area of 100,000 hm2…”. Add a reference for this statement.
- Page1: Correct “Bacillus, Pseudomonas, and Trichoderma are” by “Bacillus, Pseudomonas, and Trichoderma species are…”.
- Avoid repeating the full scientific names of species in the text. Use the abbreviation (e.g. B. subtilis, F. solani, etc.).
- Page 1: “Several studies have proven that the successful colonization of PGPR in the plant rhizosphere is the first prerequisite step [6].”. Is it several studies or one study, as you indicated at the end of the sentence?
- Page 2: Replace “a plant growth-promoting rhizobacteria” by “a PGPR”.
- Page 2: “It has been shown to suppress tomato soil-borne diseases caused by Fusarium oxysporum and Ralstonia solanacearum”. Add a reference for this statement.
- Page 2: Delete the following paragraph as it reflects the results of the study: “The dural culture test results indicated that Bacillus subtilis PTS-394 and its lipopeptides crude extract possessed excellent inhibitory activity against the causal agent of pepper root rot (Fusarium solani). Moreover, the PTS-394 wettable powder had a significant growth-promoting effect on pepper plants in the pot experiments, with its biocontrol effect on root rot reaching 69.63% at 100-fold dilution doses. Subsequent field experiments indicated that Bacillus subtilis PTS-394 wettable powder had an excellent effect at both 100- and 200-fold dilution doses and its biocontrol efficiency ranged from 68.57% to 74.43%”.
Materials and methods
- Page 7: “Fusarium solani HALJ3-1, were isolated…”. Indicate the source of the fungus (isolated from…?).
- Page 7: “Yang-zhou Luyuan Biochemical Co., Ltd”. Indicate the city and country of the company.
- Page 7: Give the full name of “YPG medium”.
- Page 7: Give the full name of “PDA medium”.
- Page 7: “in methanol and subsequent filtration (0.22 m, Nylon)”. 0.22 m or 0.22 µm? Please correct.
- Page 7: “The antifungal activity of Bacillus subtilis PTS-394 and of the lipopeptide crude extract was”. Correct the sentence.
- Page 7: “Fusarium solani HALJ3-1 was grown on PDA plates at 28°C for three days”. The fungus was grown in darkness or under light (Photoperiod)?
- Page 8: “Bruker Daltonik Reflex MALDI-TOF instrument”. Indicate the name of the company, city, and country.
- Page 8: Explain why you consider vermiculite for the soil composition.
- Page 8: “…was investigated in pot experiments. In the pot experiment, Fusarium solani HALJ3-1 was cultured…”. Delete “In the pot experiment” to avoid the repetition.
- Page 8: “a sand medium containing corn flour and sand”. Explain your interest in corn flour.
- Page 9: Correct the disease index formula as follows:
DSI (%) = [sum (class frequency × score of rating class)]/[(total number of plants) × (maximal disease index)] × 100.
Page 9: Incomprehensible sentence! “Disease index corresponded to 5 grades as follows: grade 0, the plants are not wilting; grade 1, a few leaves showing recoverable wilting; grade 3, 10% to 30% of leaves wilting, leaves are still green; grade 5, 30.1%~50% of leaves obviously wilting, leaves are yellow; grade 7, plants wilting, 50.1%~80% of leaves are yellowing and wilting, rhizome browning; grade 9, the whole plants die, roots rot”. How many disease index classes do you consider in this study? 5 or 9? Please correct.
Page 9: How many times have you repeated the treatments for the colonization experiment?
- Page 9: “field experiment was performed at a vegetable greenhouse with serious pepper root rot disease located in Huai'an city, Jiangsu Province, China”. Add the coordinates of the experiment site.
- Page 9: delete “using Excel 2019”.
- Page 9: “4.6. The colonization and biocontrol effects of Bacillus subtilis PTS-394 wettable powder in pots”. The Experimental Fields and Trial Design sections are not clear. Please rewrite two subsections (1. Colonization & 2. Biocontrol) to avoid misunderstandings.
- I have major reservations about the data analysis. Rewrite this section and specify the statistical analysis used for each treatment.
Results:
- Figure 1: According to the scale of the image, the width of the inhibition zone is not accurate. Please correct. Add a scale to each figure.
- “2.2. Identification of lipopeptide from Bacillus subtilis PTS-394 wettable powder”. Correct the subtitle as follows: “2.2. Identification of lipopeptide from Bacillus subtilis PTS-394 and its wettable powder”.
- Table 2: “The date in the table are mean±SD”. You mean the rate? Please correct.
- Page 3: Move the following sentence into the Materials & Methods Section: “The lipopeptide crude extract from PTS-394, grown in a YPG medium, and from the PTS-394 wettable powder was assessed using Matrix-assisted laser desorption/ionizationtime of flight mass spectrometry (MAL-DI-TOF-MS) in the 200-2000 nm wavelength range”.
- Figure 3: Correct the x-axis “Days after treatment with PTS-394WP” as follows: “Days after treatment”.
- Figure 3, Title. “Colonization dynamics and growth promotion of Bacillus subtilis PTS-394 in tomato pot experiments. Bacterial cell counts on pepper plant root surfaces over time following inoculation with B. subtilis PTS-394”. Colonization dynamics and growth promotion of Bacillus subtilis PTS-394 in tomato? Paper or tomato? Plagiarism! This title was copied from “Qiao J, Yu X, Liang X, Liu Y, Borriss R, Liu Y: Addition of plant-growth-promoting Bacillus subtilis PTS-394 on tomato rhizosphere has no durable impact on composition of root microbiome. BMC Microbiol 2017, 17(1):1-12”.
- Figure 3. What does it mean “FS”? Indicate this detail (Fusarium solani treatment) in the title.
- Page 5: “PTS-394G cell number”. Do you mean growth? Please correct.
- Page 5: “However, specifically on the 5th-day post application, the colonization of PTS-394 was 10-fold higher than the treatment with PTS394 (1:100) alone”. Based on the error bars (SD) as indicated in the table, this difference seems to be not statistically significant. Please clarify.
- Page 6: “indifferent”. Please correct.
- Page 6: Move the following sentence into the Materials & Methods Section: “A field experiment was performed to investigate the biocontrol efficiency of PTS-394 wettable powder on pepper root rot indifferent dilutions (1:100), (1:200), along with a control (water treated). Firstly, the PTS-394 wettable powder was mixed with the nursery soil in a 1:100 volume ratio to establish strong seedlings. After transplanting, the roots were inoculated with PTS-394 wettable powder at 1:100 and 1:200 dilutions”.
- Page 6: “was 7.96%. However, the incidence rate with PTS-394 (1:100) was 2.06%, with a biocontrol efficacy of 74.44%, and the incidence rate with PTS-394 (1:200) was 2.50%, with a biocontrol efficacy of 68.57%”. Add the SD values for each rate.
- Table 3: “The date in the table are mean±SD”. You mean the rate? Please correct.
Discussion:
- The authors did not compare their results in depth with others reported in the literature. I strongly recommend rewriting the discussion section to better understand and express the significance of the findings in relation to what was previously known.
- Page 6: “Bacillus species have been developed as biofertilizers and biopesticides due to their ability to control soil-borne diseases and promote plant growth”. Add a reference for this statement.
- Page 6: Replace “NRPS”? with “Non-ribosomal Peptide Synthetase Gene (NRPS)”.
- Page 6: Stick to either “YPG fermentation culture” or “YPG medium culture” through the text.
- Page 6: “In this study, the MALDI-TOF-MS results showed that the lipopeptides Surfactin, Iturin, and Fengycin were detected in both the YPG fermentation culture and the PTS-394 wettable powder crude extract” & “In addition, it was reported that the three major lipopeptides, Surfactin, Iturin, and Fengycin, have variable antagonistic spectrums to different fungal genera”. Merge these two sentences into one.
- Page 6: “Therefore, we proposal…”. Please correct.
- Page 6: “competence of Bacillus on the rhizosphere is…”. You mean B. subtilis?
- Page 7: “Fusarium solani”. Put the italic form in place for scientific names.
- Page 7: Replace “Previous studies have also…” by “A previous study has also…” as you refer to one paper ([20]).
- Page 7: “in Jiangsu province, China”. Delete “China” as it is supposed to indicate the name of the country in the Introduction Section (see my comment above).
- Page 7: Replace “plant growth-promoting bacteria” with “PGPR”.
- Many "key" and recent references were missed in the discussion section:
o Huang, Y.; Wu, Z.; He, Y.; Ye, B.-C.; Li, C. Rhizospheric Bacillus Subtilis Exhibits Biocontrol Effect against Rhizoctonia Solani in Pepper (Capsicum Annuum). BioMed Res. Int. 2017, 2017, e9397619, doi:10.1155/2017/9397619.
o Yi, Y.; Luan, P.; Liu, S.; Shan, Y.; Hou, Z.; Zhao, S.; Jia, S.; Li, R. Efficacy of Bacillus Subtilis XZ18-3 as a Biocontrol Agent against Rhizoctonia Cerealis on Wheat. Agriculture 2022, 12, 258, doi:10.3390/agriculture12020258.
o Wu, Z.; Huang, Y.; Li, Y.; Dong, J.; Liu, X.; Li, C. Biocontrol of Rhizoctonia Solani via Induction of the Defense Mechanism and Antimicrobial Compounds Produced by Bacillus Subtilis SL-44 on Pepper (Capsicum Annuum L.). Front. Microbiol. 2019, 10.
o Ku, Y.; Yang, N.; Pu, P.; Mei, X.; Cao, L.; Yang, X.; Cao, C. Biocontrol Mechanism of Bacillus Subtilis C3 Against Bulb Rot Disease in Fritillaria Taipaiensis P.Y.Li. Front. Microbiol. 2021, 12.
o Cavaglieri, L.; Orlando, J.; Rodríguez, M.I.; Chulze, S.; Etcheverry, M. Biocontrol of Bacillus Subtilis against Fusarium Verticillioides in Vitro and at the Maize Root Level. Res. Microbiol. 2005, 156, 748–754, doi:10.1016/j.resmic.2005.03.001.
o Zhang, D.; Qiang, R.; Zhou, Z.; Pan, Y.; Yu, S.; Yuan, W.; Cheng, J.; Wang, J.; Zhao, D.; Zhu, J.; et al. Biocontrol and Action Mechanism of Bacillus Subtilis Lipopeptides’ Fengycins Against Alternaria Solani in Potato as Assessed by a Transcriptome Analysis. Front. Microbiol. 2022, 13, 861113, doi:10.3389/fmicb.2022.861113.
o Khan, N.; Martínez-Hidalgo, P.; Ice, T.A.; Maymon, M.; Humm, E.A.; Nejat, N.; Sanders, E.R.; Kaplan, D.; Hirsch, A.M. Antifungal Activity of Bacillus Species Against Fusarium and Analysis of the Potential Mechanisms Used in Biocontrol. Front. Microbiol. 2018, 9, 2363, doi:10.3389/fmicb.2018.02363.
o Zhao, Y.; Selvaraj, J.N.; Xing, F.; Zhou, L.; Wang, Y.; Song, H.; Tan, X.; Sun, L.; Sangare, L.; Folly, Y.M.E.; et al. Antagonistic Action of Bacillus Subtilis Strain SG6 on Fusarium Graminearum. PLOS ONE 2014, 9, e92486, doi:10.1371/journal.pone.0092486.
o Wekesa, T.B.; Wekesa, V.W.; Onguso, J.M.; Wafula, E.N.; Kavesu, N. Isolation and Characterization of Bacillus Velezensis from Lake Bogoria as a Potential Biocontrol of Fusarium Solani in Phaseolus Vulgaris L. Bacteria 2022, 1, 279–293, doi:10.3390/bacteria1040021.
o Ajilogba, C.F.; Babalola, O.O.; Ahmad, F. Antagonistic Effects of Bacillus Species in Biocontrol of Tomato Fusarium Wilt. Stud. Ethno-Med. 2013, 7, 205–216, doi:10.1080/09735070.2013.11886462.
o Ju, R.; Zhao, Y.; Li, J.; Jiang, H.; Liu, P.; Yang, T.; Bao, Z.; Zhou, B.; Zhou, X.; Liu, X. Identification and Evaluation of a Potential Biocontrol Agent, Bacillus Subtilis, against Fusarium Sp. in Apple Seedlings. Ann. Microbiol. 2014, 64, 377–383, doi:10.1007/s13213-013-0672-3.
o Saha, D.; Purkayastha, G.D.; Ghosh, A.; Isha, M.; Saha, A. Isolation and Characterization of Two New Bacillus Subtilis Strains from the Rhizosphere of Eggplant as Potential Biocontrol Agents. J. Plant Pathol. 2012, 94, 109–118.
Conclusion:
Make a separate section for the conclusion.
Author Response
Comments and Suggestions for Authors
Manuscript ID: pathogens-2146606
The paper entitled “Evaluation of the Biocontrol efficiency of Bacillus subtilis wettable powder on Pepper Root Rot caused by Fusarium solani” was carefully reviewed. The purpose of this paper is to assess the biocontrol efficiency of B. subtilis PTS-394 wettable powder on pepper root rot in pot experiments and field trials.
This manuscript is fairly well written but needs extensive revision. Here are some critical points:
The relevance of the research has not been clearly identified and discussed; Technical clarifications: many deficiencies in the Materials & Methods section compromise the understanding of ideas and concepts and affect the quality of the paper; Poorly prepared discussion lacking appropriate interpretation (out of context!). The Discussion Section should be concise and relevant to the results. Furthermore, some statements are simply incorrect and/or are not supported by any scientific literature. Many "key" references have been missed, and the list of references is not up-to-date; The manuscript is full of typos.
Response: We would like to thank you for your comments. Your suggestion is very valuable to improve the quality of our manuscripts. As your suggestion, we have made the point-by-point revised and modified the Figures and some other typo in red.
Detailed comments:
Abstract: Improve on the conclusions of the abstract.
Response: we have rewritten the conclusion of the abstract.
Introduction:This section still needs to be deeply improved. Enlarge the state of the art by adding other relevant and recent works in the field. Add one or two paragraphs in the Introduction Section to present the studies elucidating the efficacy of Bacillus subtilis against pathogenic fungi in pepper crops.
Response: we have added the statement of recently studies of Bacillus species against pepper desease in the introduction.
Page 1: “Jiangsu Province, …”. Indicate the country.
Response: we have added the country in text.
Page 1: “with an annual planting area of 100,000 hm2…”. Add a reference for this statement.
Response: we have added a reference in text.
Page1: Correct “Bacillus, Pseudomonas, and Trichoderma are” by “Bacillus, Pseudomonas, and Trichoderma species are…”.
Response: we have corrected this sentence as your suggestion.
Avoid repeating the full scientific names of species in the text. Use the abbreviation (e.g. B. subtilis, F. solani, etc.).
Response: we have corrected the species names as abbreviation in text.
Page 1: “Several studies have proven that the successful colonization of PGPR in the plant rhizosphere is the first prerequisite step [6].”. Is it several studies or one study, as you indicated at the end of the sentence?
Response: we have added anther reference and rewritten the sentence.
Page 2: Replace “a plant growth-promoting rhizobacteria” by “a PGPR”.
Response: we have replace “a plant growth-promoting rhizobacteria” by “a PGPR”.
Page 2: “It has been shown to suppress tomato soil-borne diseases caused by Fusarium oxysporum and Ralstonia solanacearum”. Add a reference for this statement.
Response: we have added a reference for this statement.
Page 2: Delete the following paragraph as it reflects the results of the study: “The dural culture test results indicated that Bacillus subtilis PTS-394 and its lipopeptides crude extract possessed excellent inhibitory activity against the causal agent of pepper root rot (Fusarium solani). Moreover, the PTS-394 wettable powder had a significant growth-promoting effect on pepper plants in the pot experiments, with its biocontrol effect on root rot reaching 69.63% at 100-fold dilution doses. Subsequent field experiments indicated that Bacillus subtilis PTS-394 wettable powder had an excellent effect at both 100- and 200-fold dilution doses and its biocontrol efficiency ranged from 68.57% to 74.43%”.
Response: we have deleted these statements.
Materials and methods
Page 7: “Fusarium solani HALJ3-1, were isolated…”. Indicate the source of the fungus (isolated from…?).
Response: we have added the detail of Fusarium solani HALJ3-1.
Page 7: “Yang-zhou Luyuan Biochemical Co., Ltd”. Indicate the city and country of the company.
Response: we have given the location details of the company.
Page 7: Give the full name of “YPG medium”.
Response: we have given the full name of “YPG medium”.
Page 7: Give the full name of “PDA medium”.
Response: we have given the full name of “PDA medium”.
Page 7: “in methanol and subsequent filtration (0.22 m, Nylon)”. 0.22 m or 0.22 µm? Please correct.
Response: we have corrected as 0.22 µm in this section.
Page 7: “The antifungal activity of Bacillus subtilis PTS-394 and of the lipopeptide crude extract was”. Correct the sentence.
Response: we have corrected this sentence.
Page 7: “Fusarium solani HALJ3-1 was grown on PDA plates at 28°C for three days”. The fungus was grown in darkness or under light (Photoperiod)?
Response: The fungus F.solani was grown in darkness and we have added the detail.
Page 8: “Bruker Daltonik Reflex MALDI-TOF instrument”. Indicate the name of the company, city, and country.
Response: we have added details in this section.
Page 8: Explain why you consider vermiculite for the soil composition.
Response: Vermiculite is an expanded mineral produced by mica rock under the action of high temperature, which is often used in the cultivation of horticultural crops. Mixing the appropriate amount of vermiculite in the soil has the effect of improving the porosity of the soil and increasing the permeability. when watered soil thoroughly, there is not difference between the humidity of the upper and lower layers of the soil, which has both the effect of water permeability and water retention. Moreover, vermiculite is also an inorganic mineral that can stimulate plant growth and promote root growth. Therefore, the addition of vermiculite in this study is beneficial to the growth of pepper seedlings.
Page 8: “…was investigated in pot experiments. In the pot experiment, Fusarium solani HALJ3-1 was cultured…”. Delete “In the pot experiment” to avoid the repetition.
Response: we have deleted“In the pot experiment” as your suggestion.
Page 8: “a sand medium containing corn flour and sand”. Explain your interest in corn flour.
Response: Fursarium solani is a soil-born pathogen, can causes root rot of pepper. So, the pathogenic soil is a useful inoculum for the development of pepper root rot. The sand corn medium was prepared with sand and corn for culturing of Fursarium. Then, mixing the sand corn medium with the nomal soil in proportion as the pathogenic soil is used in pot experiments. Fursarium species can causes the desease of corn and grows well on corn as a nutrient. The low price and rich nutrition of corn flour are very suitable for the cultivation of Fursarium solani. Therefore, the corn flour was chosed to prepare the sand corn medium in this study.
Page 9: Correct the disease index formula as follows:
DSI (%) = [sum (class frequency × score of rating class)]/[(total number of plants) × (maximal disease index)] × 100.
Response: we have corrected the formula as your suggestion.
Page 9: Incomprehensible sentence! “Disease index corresponded to 5 grades as follows: grade 0, the plants are not wilting; grade 1, a few leaves showing recoverable wilting; grade 3, 10% to 30% of leaves wilting, leaves are still green; grade 5, 30.1%~50% of leaves obviously wilting, leaves are yellow; grade 7, plants wilting, 50.1%~80% of leaves are yellowing and wilting, rhizome browning; grade 9, the whole plants die, roots rot”. How many disease index classes do you consider in this study? 5 or 9? Please correct.
Response: we have corrected the disease index classes as 9 grades.
Page 9: How many times have you repeated the treatments for the colonization experiment?
Response: we have carried out this experiment according to three replications per treatment. The method detail was added in the method section as follow.
Pepper root samples were collected from treatments of PTS-394 WP (1:100) and PTS-394 WP (1:100) +FS at 1, 3, 5, 7, 9, 15, and 25 days after transplantation. At the time of sam-pling, a total of two pepper seedlings were collected per treatment and replicated 3 times. The soil on the root of two pepper plants was removed as clean as possible, and the root was removed with ultrasonic waves for 20 min at a ratio of 10 ml sterile water per gram of root. Afterwards, the bacterial solution was diluted 104 or 105 times and plated on YPG agar petri dish to count the number of PTS-394 colonies. Then, the amount of PTS-394 per gram of root was calculated. Statistical analysis was performed on the three replicates.
Page 9: “field experiment was performed at a vegetable greenhouse with serious pepper root rot disease located in Huai'an city, Jiangsu Province, China”. Add the coordinates of the experiment site.
Response: we have added the coordinates of the experiment site as your suggestion.
Page 9: delete “using Excel 2019”.
Response: we have deleted “using Excel 2019” as your suggestion.
Page 9: “4.6. The colonization and biocontrol effects of Bacillus subtilis PTS-394 wettable powder in pots”. The Experimental Fields and Trial Design sections are not clear. Please rewrite two subsections (1. Colonization & 2. Biocontrol) to avoid misunderstandings.
Response: we have rewritten this section into two sections as your suggestion.
I have major reservations about the data analysis. Rewrite this section and specify the statistical analysis used for each treatment.
Response: we have rewritten this section and presented the specific statistical anaysis for each treatment.
Results:
Figure 1: According to the scale of the image, the width of the inhibition zone is not accurate. Please correct. Add a scale to each figure.
Response: In order to obvsered clearly, we have added a scale to each figure in the bottom of Figure; in the meantime, the red arrows and yellow circles were marked.
“2.2. Identification of lipopeptide from Bacillus subtilis PTS-394 wettable powder”. Correct the subtitle as follows: “2.2. Identification of lipopeptide from Bacillus subtilis PTS-394 and its wettable powder”.
Response: we have corrected the statement as your suggetion.
Table 2: “The date in the table are mean±SD”. You mean the rate? Please correct.
Response: I am sorry for my typos mistake, Here should be “The data in the table are mean±SD” not the “date”.
Page 3: Move the following sentence into the Materials & Methods Section: “The lipopeptide crude extract from PTS-394, grown in a YPG medium, and from the PTS-394 wettable powder was assessed using Matrix-assisted laser desorption/ionizationtime of flight mass spectrometry (MAL-DI-TOF-MS) in the 200-2000 nm wavelength range”.
Response: we have moved the sentence into the Materials & Methods Section, and rewritten the statement for understand easily.
Figure 3: Correct the x-axis “Days after treatment with PTS-394WP” as follows: “Days after treatment”.
Response: we have corrected the x-axis as your suggetion.
Figure 3, Title. “Colonization dynamics and growth promotion of Bacillus subtilis PTS-394 in tomato pot experiments. Bacterial cell counts on pepper plant root surfaces over time following inoculation with B. subtilis PTS-394”. Colonization dynamics and growth promotion of Bacillus subtilis PTS-394 in tomato? Paper or tomato? Plagiarism! This title was copied from “Qiao J, Yu X, Liang X, Liu Y, Borriss R, Liu Y: Addition of plant-growth-promoting Bacillus subtilis PTS-394 on tomato rhizosphere has no durable impact on composition of root microbiome. BMC Microbiol 2017, 17(1):1-12”.
Response: we have check the Figure 3 title, and revised in the text.
Figure 3. What does it mean “FS”? Indicate this detail (Fusarium solani treatment) in the title.
Response:“FS” means Fusarium solani treatment, we have corrected and added detail.
Page 5: “PTS-394G cell number”. Do you mean growth? Please correct.
Response: I am sorry for my typos mistake, we have corrected as “PTS-394 cell number”.
Page 5: “However, specifically on the 5th-day post application, the colonization of PTS-394 was 10-fold higher than the treatment with PTS394 (1:100) alone”. Based on the error bars (SD) as indicated in the table, this difference seems to be not statistically significant. Please clarify.
Response: I am appreciative of the your carefully analysis of the data. We have checked the original data and noticed that there was an typo in the statistical analysis of the data at day 5. We performed the data statistics again. Independent two-sample t-test was performed in the significantly analysis of the dynamic colonization of B. subtilis PTS-394. The results showed that the amount of colonization of PTS-394 in the PTS-394 WP (1:100) +FS treatment kept significantly higher than that of the treatment PTS-394 at 3, 5 and 7 days after inoculation. The new results has added in the manuscript and the Fig 3 was replaced too.
Page 6: “indifferent”. Please correct.
Response: we have corrected as “in different”.
Page 6: Move the following sentence into the Materials & Methods Section: “A field experiment was performed to investigate the biocontrol efficiency of PTS-394 wettable powder on pepper root rot indifferent dilutions (1:100), (1:200), along with a control (water treated). Firstly, the PTS-394 wettable powder was mixed with the nursery soil in a 1:100 volume ratio to establish strong seedlings. After transplanting, the roots were inoculated with PTS-394 wettable powder at 1:100 and 1:200 dilutions”.
Response: we have moved the statement into the Materials & Methods Section, and re-organization the method for understand easily.
Page 6: “was 7.96%. However, the incidence rate with PTS-394 (1:100) was 2.06%, with a biocontrol efficacy of 74.44%, and the incidence rate with PTS-394 (1:200) was 2.50%, with a biocontrol efficacy of 68.57%”. Add the SD values for each rate.
Response: we have added SD values for each data.
Table 3: “The date in the table are mean±SD”. You mean the rate? Please correct.
Response: Here should be “The data in the table are mean±SD” not the “date”.
Discussion:
The authors did not compare their results in depth with others reported in the literature. I strongly recommend rewriting the discussion section to better understand and express the significance of the findings in relation to what was previously known.
Response: we have read the key and recent new references as your suggetion and the discussion has been rewritten.
Page 6: “Bacillus species have been developed as biofertilizers and biopesticides due to their ability to control soil-borne diseases and promote plant growth”. Add a reference for this statement.
Response: we have added a reference for this statement.
Page 6: Replace “NRPS”? with “Non-ribosomal Peptide Synthetase Gene (NRPS)”.
Response: we have replaced “NRPS”? with “Non-ribosomal Peptide Synthetase Gene (NRPS)”.
Page 6: Stick to either “YPG fermentation culture” or “YPG medium culture” through the text.
Response: we have revised “YPG fermentation culture” as“YPG medium culture” in the whole text.
Page 6: “In this study, the MALDI-TOF-MS results showed that the lipopeptides Surfactin, Iturin, and Fengycin were detected in both the YPG fermentation culture and the PTS-394 wettable powder crude extract” & “In addition, it was reported that the three major lipopeptides, Surfactin, Iturin, and Fengycin, have variable antagonistic spectrums to different fungal genera”. Merge these two sentences into one.
Response: we have combined two sentenced into one as follow: “In this study, the MALDI-TOF-MS results showed that the lipopeptides Surfactin, Iturin, and Fengycin were detected in both the YPG fermentation culture and the PTS-394 wettable powder crude extract, which have variable antagonistic spectrums to different fungal genera”.
Page 6: “Therefore, we proposal…”. Please correct.
Response: we have corrected as “Therefore, we proposed….”
Page 6: “competence of Bacillus on the rhizosphere is…”. You mean B. subtilis?
Response: we have revised “Bacillus” as Bacillus species.
Page 7: “Fusarium solani”. Put the italic form in place for scientific names.
Response: we have revised the word “Fusarium solani” as italic.
Page 7: Replace “Previous studies have also…” by “A previous study has also…” as you refer to one paper ([20]).
Response: we have replaced “Previous studies have also…” by “A previous study has also…”.
Page 7: “in Jiangsu province, China”. Delete “China” as it is supposed to indicate the name of the country in the Introduction Section (see my comment above).
Response: we have deleted “China” as your suggestion.
Page 7: Replace “plant growth-promoting bacteria” with “PGPR”.
Response: we have replaced“plant growth-promoting bacteria” with “PGPR”.
Many "key" and recent references were missed in the discussion section:
Huang, Y.; Wu, Z.; He, Y.; Ye, B.-C.; Li, C. Rhizospheric Bacillus Subtilis Exhibits Biocontrol Effect against Rhizoctonia Solani in Pepper (Capsicum Annuum). BioMed Res. Int. 2017, 2017, e9397619, doi:10.1155/2017/9397619.
Yi, Y.; Luan, P.; Liu, S.; Shan, Y.; Hou, Z.; Zhao, S.; Jia, S.; Li, R. Efficacy of Bacillus Subtilis XZ18-3 as a Biocontrol Agent against Rhizoctonia Cerealis on Wheat. Agriculture 2022, 12, 258, doi:10.3390/agriculture12020258.
Wu, Z.; Huang, Y.; Li, Y.; Dong, J.; Liu, X.; Li, C. Biocontrol of Rhizoctonia Solani via Induction of the Defense Mechanism and Antimicrobial Compounds Produced by Bacillus Subtilis SL-44 on Pepper (Capsicum Annuum L.). Front. Microbiol. 2019, 10.
Ku, Y.; Yang, N.; Pu, P.; Mei, X.; Cao, L.; Yang, X.; Cao, C. Biocontrol Mechanism of Bacillus Subtilis C3 Against Bulb Rot Disease in Fritillaria Taipaiensis P.Y.Li. Front. Microbiol. 2021, 12.
Cavaglieri, L.; Orlando, J.; Rodríguez, M.I.; Chulze, S.; Etcheverry, M. Biocontrol of Bacillus Subtilis against Fusarium Verticillioides in Vitro and at the Maize Root Level. Res. Microbiol. 2005, 156, 748–754, doi:10.1016/j.resmic.2005.03.001.
Zhang, D.; Qiang, R.; Zhou, Z.; Pan, Y.; Yu, S.; Yuan, W.; Cheng, J.; Wang, J.; Zhao, D.; Zhu, J.; et al. Biocontrol and Action Mechanism of Bacillus Subtilis Lipopeptides’ Fengycins Against Alternaria Solani in Potato as Assessed by a Transcriptome Analysis. Front. Microbiol. 2022, 13, 861113, doi:10.3389/fmicb.2022.861113.
Khan, N.; Martínez-Hidalgo, P.; Ice, T.A.; Maymon, M.; Humm, E.A.; Nejat, N.; Sanders, E.R.; Kaplan, D.; Hirsch, A.M. Antifungal Activity of Bacillus Species Against Fusarium and Analysis of the Potential Mechanisms Used in Biocontrol. Front. Microbiol. 2018, 9, 2363, doi:10.3389/fmicb.2018.02363.
Zhao, Y.; Selvaraj, J.N.; Xing, F.; Zhou, L.; Wang, Y.; Song, H.; Tan, X.; Sun, L.; Sangare, L.; Folly, Y.M.E.; et al. Antagonistic Action of Bacillus Subtilis Strain SG6 on Fusarium Graminearum. PLOS ONE 2014, 9, e92486, doi:10.1371/journal.pone.0092486.
Wekesa, T.B.; Wekesa, V.W.; Onguso, J.M.; Wafula, E.N.; Kavesu, N. Isolation and Characterization of Bacillus Velezensis from Lake Bogoria as a Potential Biocontrol of Fusarium Solani in Phaseolus Vulgaris L. Bacteria 2022, 1, 279–293, doi:10.3390/bacteria1040021.
Ajilogba, C.F.; Babalola, O.O.; Ahmad, F. Antagonistic Effects of Bacillus Species in Biocontrol of Tomato Fusarium Wilt. Stud. Ethno-Med. 2013, 7, 205–216, doi:10.1080/09735070.2013.11886462.
Ju, R.; Zhao, Y.; Li, J.; Jiang, H.; Liu, P.; Yang, T.; Bao, Z.; Zhou, B.; Zhou, X.; Liu, X. Identification and Evaluation of a Potential Biocontrol Agent, Bacillus Subtilis, against Fusarium Sp. in Apple Seedlings. Ann. Microbiol. 2014, 64, 377–383, doi:10.1007/s13213-013-0672-3.
Saha, D.; Purkayastha, G.D.; Ghosh, A.; Isha, M.; Saha, A. Isolation and Characterization of Two New Bacillus Subtilis Strains from the Rhizosphere of Eggplant as Potential Biocontrol Agents. J. Plant Pathol. 2012, 94, 109–118.
Response: we have rewritten the discusion and many key and recent references was cited in the literture list.
Conclusion:
Make a separate section for the conclusion.
Response: we have revised the conclusion as separate section.

Round 2
Reviewer 1 Report
Dear authors you have almost revised your manuscript appropriately according to the suggestions provided and but still, some changes are required
Comment 1: The disc diffusion method described in the revised manuscript must be entitled with appropriate refrence to any research paper that have used the method.
Comment 2: Fig 1 as mentioned in the previous revision has been suggested to be changed but the figure has not been changed but same figure has been modified into sections. The current plates of Control and treated do not visually show much difference.
Author Response
Comments and Suggestions for Authors
Dear authors you have almost revised your manuscript appropriately according to the suggestions provided and but still, some changes are required
Response: Thanks for your suggestion. We had revised the manuscript according to your specific comments and labled in red.
Comment 1: The disc diffusion method described in the revised manuscript must be entitled with appropriate refrence to any research paper that have used the method.
Response: We had added the references about the disc diffusion method.
Comment 2: Fig 1 as mentioned in the previous revision has been suggested to be changed but the figure has not been changed but same figure has been modified into sections. The current plates of Control and treated do not visually show much difference.
Response:The Fig1 was changed and the control plate was added.

Reviewer 3 Report
The authors have done extensive research on the topic and have presented interesting work in the revised version of the manuscript. They have responded to all comments.
Author Response
Comments and Suggestions for Authors
The authors have done extensive research on the topic and have presented interesting work in the revised version of the manuscript. They have responded to all comments.
Response: Thank you for your comments which are very useful to improve the quality of our manuscript. Thank you again.